# Granular Resistive Force Theory Extension for Saturated Wet Sand Ground

**Xinmeng Ma [1,2]**, **Gang Wang [2,*]**, **Kaixin Liu [2]**, **Xi Chen [3,4]**, **Jixin Wang [1]**, **Biye Pan [2]** and **Liquan Wang [1]**

[1]  College of Mechanical Electronical and Engineering, Harbin Engineering University, Harbin 150001, China
[2]  Science and Technology on Underwater Vehicle Laboratory, Harbin Engineering University, Harbin 150001, China
[3]  College of Information and Communication Engineering, Harbin Engineering University, Harbin 150001, China
[4]  College of Mechanical and Electrical Engineering, Heilongjiang Institute of Technology, Harbin 150050, China
*  Correspondence: wanggang@hrbeu.edu.cn; Tel.: +86-0451-82568056

**Abstract:** Amphibious environments formed from sand and water present a formidable challenge to the running motion of field robots, as the mixing of granular media (GM) and water makes the force laws of robotic legs more complicated during robot running. To this end, we extended the granular resistive force theory (RFT) to saturated wet granular media, named saturated granular RFT (SGRFT), which can be suitable for saturated wet sand submerged in water. This method can extend RFT for dry GM to saturated wet granular media (SWGM) by using the method's velocity and depth coefficient. The force laws of the robotic legs in dry GM and SWGM were tested, compared, and analyzed. The difference in force laws between the two kinds of media, from the sensitivity to speed (10 mm/s~50 mm/s) and depth (0~60 mm), was calculated. More than 70% of the prediction results of the horizontal resistive force using SGRFT have an error of less than 6%. The effectiveness of the SGRFT in legged robots is proved by simulation and testing of three kinds of legs. The difference in force laws when running is proved by the experiments of the robot equipped with the propeller-leg in dry GM and SWGM, which is vital for amphibious robots working in shoal environments (including dry GM and SWGM ground).

**Keywords:** contact modeling; field robots; legged locomotion; wet granular media

## 1. Introduction

The ground formed by granular media occupies a large area, such as the beach and seabed encountered by amphibious robots and even the surface of Mars, which is encountered in interstellar exploration. The special force laws of granular media that differ from solids and fluids influence the theory of mobility in solid ground. Conventional fluids do not solve the problems robots encounter on the surface of granular media. The unfortunate instance of NASA's Spirit rover becoming immobilized on Mars proves the challenge [1], as the ground materials Spirit encountered were GM. The resistive force theory for GM was studied in this case, the challenge was solved, and the athletic ability of NASA's rover was dramatically improved [2,3]. The way the field robot interacts with the granular media in the working process plays a vital role in the motion performance of the robot [4]. The ground of beach and seabed formed by sand and water is SWGM. The force laws for SWGM are more complex than those of GM, so it is valuable to study the resistive force theory of SWGM.

The robot–soil interaction models and granular media resistive force theory (RFT) are effective methods for analyzing and predicting a robot's condition when running in the field environment. The robot–soil interaction models indicate the robot's behavior in yielding terrain, relying on the parameters tested by the sinkage and shear tests [5–7]. Still, the gaps between observed and estimated performances are inevitable due to the

uncertainty in the terrain parameter estimation and the interaction models. Granular media resistive force theory (RFT) utilizes superposition and discretization of intruders into smaller geometries to sum the resultant forces for analysis [8,9]. A simplified RFT for dry, uniform granular materials in horizontal motion planes has been developed. It has been demonstrated that resistive forces are scaled with the cross-sectional area and are mainly independent of intrusion speed at low speeds ($\leq 1$ m/s) [10]. The studies in another article also proved its effectiveness [11]. Granular scaling laws (GSL) represent another method entailing the direct scaling of various parameters such as size and mass, where certain outputs such as velocity and power for larger wheels of the same general shape can be predicted from smaller ones [12]. It is a feasible idea to solve the problems encountered by robots on the granular media ground by simplifying the model and calculating results using dimensionless parameters.

There have been many successful cases in the study of the force laws between robots and granular media to optimize the movement abilities of robots. The mobility method of snakes has been observed, and the force law on the slope has been studied with dry granular media [13]. Treers et al. implemented the RTF for three-dimensional trajectories on dry granular media, called 3D RTF [14]. A dynamic resistive force model (DRFM) method suitable for 3D dynamic intrusion was introduced, and a complete set of solutions was proposed to design mobile robot wheels in granular media [15]. Agarwal et al. showed that a continuum model based on the simple notions of frictional flow and tension-free separation describes complex granular intrusions near free surfaces. They merged these effects into a further reduced-order technique (dynamic resistive force theory) for rapid modeling of the granular locomotion of arbitrarily shaped intruders [3].

In recent years, the force theory of soft robots on granular media ground has been studied. The design of soft digging robots inspired by the bristled worm has been investigated. Two methods of actuation for the leading segment have been implemented and studied: periodic radial expansion and bidirectional bending [16]. Ortiz et al. studied the mechanism of entering the dry granular media and realized the movement of the soft robot in the dry granular media [17]. Li et al. studied the behavior of soft fins in granular media to predict the time-based evolution of soft mechanisms [18].

Adding a new strategy that relies on the force theory to the robot can improve its working ability. Shrivastava et al. showed a peculiar gait strategy that agitates and cyclically reflows grains under the robot, allowing it to "swim" up loosely consolidated hills [2]. Fernandez et al. improved the dragging ability of wheeled robots on dry granular media by anchoring, which shows that in addition to studying the mechanism, it is also important to adopt appropriate strategies [19]. Lee et al. optimized the spoke-based wheels called DODO that run on granular media using the Taguchi method and L9 orthogonal array [20].

There has been much research on mobile robots in granular media. Still, the current research conclusions are not enough to predict legged robot mobility on wet granular media ground. The research summarized above concerns dry granular media (GM). The resistive force on the legs (the reaction force of the ground to the robot) is mainly provided by the friction between the legs and the GM, and the inertial force of the GM. The friction force plays a dominant role at low speed, and the effect of the inertial force increases significantly at high speed [3]. Thus, the effect of velocity on force is usually ignored at low velocity ($\leq 1$ m/s) [10]. For amphibious robots running in shoal environments, wet granular media (SWGM) ground covered by water is one of the most common ground environments. SWGM is made up of a mixture of sand and water. The force laws become more complicated when the gaps between the particles are filled with water. The RFT for GM is not enough to solve the problem that robots face in SWGM. Sharpe et al. studied the force law of the wet granular media with water content preparations of 1%, 3%, and 5%. They observed a lizard's movement behavior through X-ray, letting it move in the wet sand and analyzing its force. The study proves that the resistance in wet sand is about four times higher than that in dry sand, indicating significant differences in rheology between dry sand and wet sand. It expanded the research from a single media to a multimedia

mixture [21]. Although this study points out that the force laws in GM and SWGM are quite different, it does not give a usable theoretical model that can be applied to predict robot mobility.

The movement mechanism of organisms on seabed wet sand has been studied and published. The peculiar burrowing behavior of sand lances was studied, and the roles of slipping wave and non-slipping wave locomotion during burrowing in the sand were determined by standard and X-ray video. It was proved that as soon as enough of the body is underground, sand lances undergo a kinematic shift and locomote like limbless terrestrial vertebrates [22]. Winter and Hosoi et al. studied the Atlantic razor clam's behavior when it burrows into ocean substrates. Its low-energy requirements by localized fluidization are helpful for anchoring, installing cables, and recovering oil [23–27].

Other alternatives are potentially suitable to predict the force interactions between particles and compliant intruders. Discrete element method (DEM) [28] coupled with finite element method (FEM)-based simulations [29–31] have been proven to be effective. However, the high computational costs reduce their usefulness in optimization-based approaches.

In this work, we adopted the idea of RFT to study the force laws of robot legs when running on saturated wet granular media (sea sand, particle diameter $\leq 1$ mm) submerged in water. The research in this paper renders the RFT in GM suitable for SWGM, and we can predict the resistive forces on the legs as they run in the SWGM, only needing to add two coefficients that can be obtained by testing. Firstly, we built a force measurement platform and analyzed the similarities and differences in force laws between wet granular media and dry granular media. We measured and analyzed by observing the behavior of the intruder (a small aluminum alloy plate 20 mm $\times$ 20 mm $\times$ 2 mm) moving in the granular media at different speeds and depths. The test results show that the force change in the wet granular media is more sensitive to speed. Secondly, according to the test results, we added the velocity coefficient to the RFT of GM and adjusted the depth coefficient to obtain the RFT in SWGM (SGRFT). We obtained the velocity coefficient and depth coefficient when the velocity is less than 50 mm/s, and the validity of the coefficients is proved by the prediction of a group of horizontal resistive forces. Through the simulation and testing of the resistive force of three legs with different shapes on the SWGM ground, it is proven that the SGRFT can be applied to the motion of legged robots. The resistive force of the three shapes of legs at different speeds was measured, which proved the effectiveness of the coefficients. Finally, the comparative experiment of the robot running on the ground of GM and SWGM proves the force law in this paper. Specific information about the robot is explained in our previous study [32].

## 2. Method and Materials

The measurement and comparison of GM and SWGM consist of five main steps.

### 2.1. STEP 1: Equipment Preparation

We used the test setup illustrated in Figure 1 to measure force and control displacement in tests. The mounting side of a 6-DOF force/torque sensor (SRI-M4325k1) was connected to the end of the spindle of a three-axis machine center (Ningbo KAIBO CNC, DC-1317). The intruders were mounted to the tool side of the sensor with a custom attachment in various tests. The intruders were inserted into a box filled with sea sand. The particle diameter of sand sieved by 120 mesh screens was $\leq 1$ mm. The attachment for connecting intruders and sensors is shown in Figure 1. It is required to measure the force of the attachment before making a new measurement that the speed or depth is changed. The force of the intruder is determined by the total force measured minus the force of the attachment. The details of the equipment are shown in Table 1.

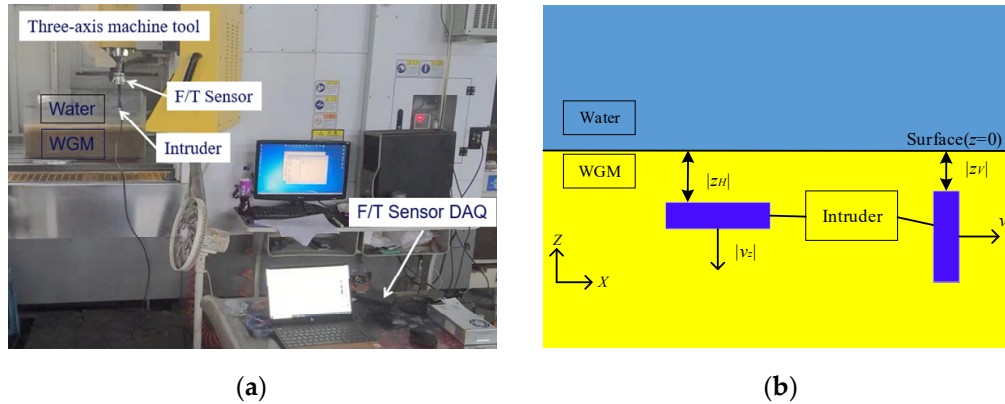

**(a)** **(b)**

**Figure 1.** The test setup. (**a**) The test setup was used in this study. Including a three-axis machining center, an SRI force/torque sensor with DAQ, and a box of wet granular media (SWGM). The intruder is attached to the test setup using a custom metal plate attachment and inserted into the box placed on the test table. (**b**) The horizontal and vertical motion states of intruders in SWGM and the definition of depth and speed. The machining center records the location of the intruder. The speed of the intruder is the speed of the machine tool.

**Table 1.** The details of the test equipment.

|  | Equipment | Model or Material | Key Parameters |
|---|---|---|---|
| 1 | Machine center | NINGBO KAIBO CNC, DC-1317 | Position accuracy: $\leq$0.01 mm |
| 2 | F/T sensor | SRI-M4325k1 | Range: Fx = Fy = Fz $\leq$ 200 N, Mx = My = Mz $\leq$ 20 Nm |
| 3 | Intruder | Aluminum alloy 7075 | Projected area in the direction of $v$: 20 mm $\times$ 20 mm<br>Thickness: 2 mm |
| 5 | Granular media | Sea sand | Particle diameter: $\leq$1 mm |
| 6 | Box | Plexiglass | Length $\times$ width $\times$ height = 800 mm $\times$ 800 mm $\times$ 400 mm |

### 2.2. STEP 2: Key Parameter Measurement

Before each test, we put 125 kg of sand in the box and scraped it to the marking line 100 mm from the bottom of the box with a scraper to ensure that the volume fraction, $\phi$, of each test was the same, which is 0.73, the definition of $\phi$ is as in (1). Before the tests of wet granular media, we added water to the box until the water surface was just above the sand surface, and calculated the $W$ by (2), where $W$ is the water content. The moisture content of the three tests was 21.6%, 21%, and 21.5%, which proves that there is an approximately fixed saturated moisture content under the condition of the same type of sand and the same filling rate. The arithmetic average of the three measurement results is about 21.4%. After reaching this moisture content in each test, water continued to be injected until the water surface was 50 mm higher than the sand surface. The volume fraction ($\phi$), water content ($W$), and bulk volume of GM ($V_{\text{GM}}$) are calculated by Equations (1)–(3).

$$\phi = V_{\text{a}} / V_{\text{GM}}, \tag{1}$$

$$W = m_{\text{water}} / m_{\text{GM}}, \tag{2}$$

$$V_{\text{GM}} = M_{\text{wet}} / [\rho_{\text{GM}}(1 + W)], \tag{3}$$

where $\rho_{\text{GM}}$ is the density of the GM, $M_{\text{wet}}$ is the mass of the wet media, $V_{\text{a}}$ is the absolute compact volume of GM, $V_{\text{GM}}$ is the bulk volume of GM, $m_{\text{water}}$ is the quality of the water added, and $m_{\text{GM}}$ is the quality of the GM.

### 2.3. STEP 3: Two Calibrations for the Test

The measured force result will be affected by the near-wall effect if the intruder is closer to the box wall during the test, so it is necessary to calibrate the effective distance. It

was proved in [10] that this effect can be avoided when the distance from the side wall is more than 30 mm during the test of dry granular media. Before the wet granular media tests, the intruder was measured from the position of 10 mm to 70 mm from the side wall 7 times, and the distance between the position of test(n) and the position of test(n + 1) was 10 mm. The measuring depth was 10 mm, and the speed was 30 mm/s. The measurement results are shown in Figure 2a. The measurement results proved that when testing in wet granular media, if the distance from the wall is more than 40 mm, it is not affected by the near-wall effect, and the measurement result is credible.

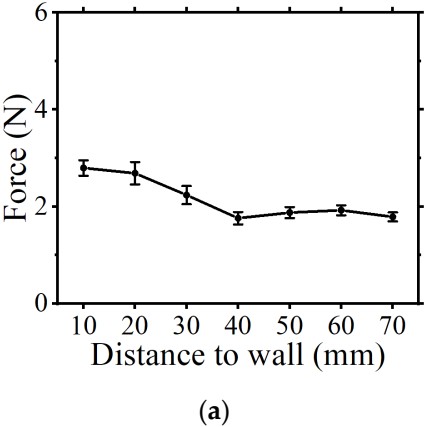
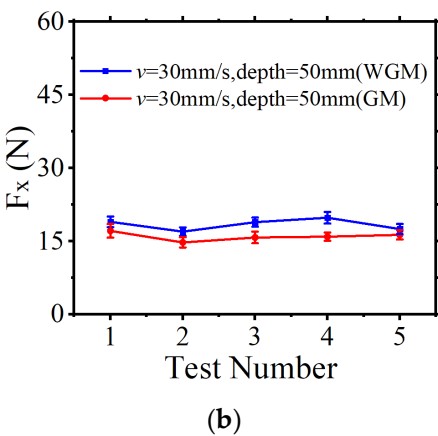

(**a**)  (**b**)

**Figure 2.** The results of calibrations. (**a**) The measurement results of the wall effect for SWGM. The force is no longer affected by the wall when the intruder is more than 40 mm away from the wall. (**b**) The outcome of the consistency test. The results on five prepared media under the same exercise conditions demonstrate the consistency of the test conditions. Error bars are standard deviations, with three trials for each condition.

To ensure the consistency of the tests, we added the same quality of sand into the box and scraped the sand to the same height each time with a scraper. We tested the effectiveness of this method through repeated trials. We carried out five tests with the same parameters; we dragged the intruder horizontally at a depth of 50 mm at a speed of 30 mm/s for three seconds, and after each test, the wet granular media was evenly stirred and then re-scraped. The wet granular media that was scraped, the definition of depth, and the test results are shown in Figure 2b. The statistical results in Figure 2 prove that this method can ensure the consistency of the test conditions.

### 2.4. STEP 4: Tests of the Horizontal Force

The horizontal force measurement results are shown in Figure 3, including dry granular media and wet granular media. Figure 3a shows the change in horizontal force with speed in two kinds of media. The test depths are 0 mm, 30 mm, and 50 mm, and the test speed is from 10 mm/s to 50 mm/s; the speed increases by 10 mm/s between the two tests.

It can be seen from the test results that the phenomenon observed in the dry granular media is the same as that described in the previous study [10]. The horizontal force in the test is not sensitive to speed, the force did not show a significant change when the speed changed by 5-fold times, and the maximum difference was only 5%.

As shown in Figure 3a, the sensitivity of the horizontal force in wet granular media to speed is significantly higher than that in dry granular media. At a depth of 30 mm, the horizontal force when moving at a speed of 50 mm/s is 178.6% of that when moving at a speed of 10 mm/s. It can be seen from Figure 3 that the horizontal force in the wet granular media in the low-speed range (10 mm/s~20 mm/s) is lower than that in the dry granular media. When the speed is higher than 30 mm/s, the horizontal force in the wet granular media is higher than that in the dry granular media. The force in SWGM increases significantly with increasing speed. This performance is more similar to the corresponding relationship between force and speed in the fluid, but it is not linearly related to the square

of the speed. This shows the characteristics of wet granular media, which are different from fluids and solids and significantly different from dry granular media.

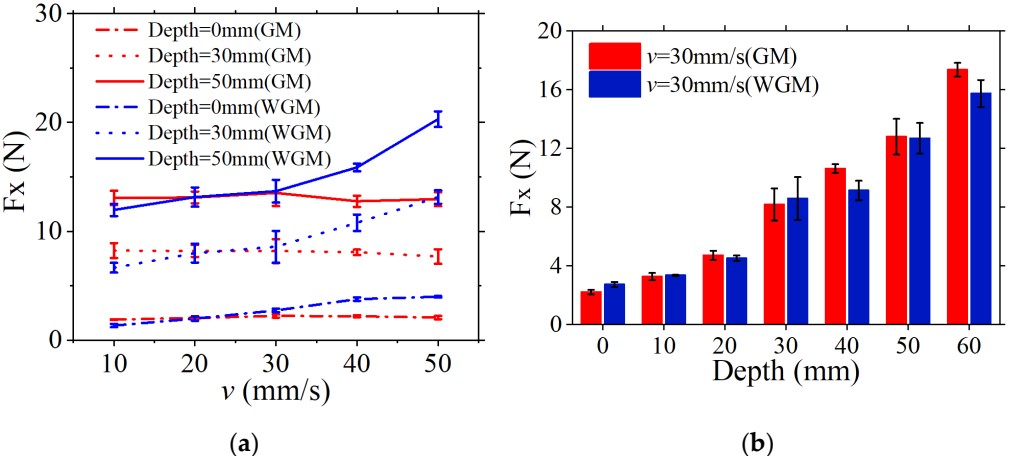

**Figure 3.** The results of the horizontal force test. (**a**) Horizontal force results of 15 groups of comparative tests (three depths, five speeds). The variation trend of force with velocity in the two media at the same depth is different. (**b**) Horizontal force results of 7 groups of comparative tests. The variation trend of force with depth in both media is approximately linear. Error bars are standard deviations, with three trials for each condition.

Figure 3b shows the change in horizontal force with depth in the two kinds of media. The test speed is 30 mm/s. The previous research results in Figure 2 show that the force of the intruder in the two types of media at this speed is the closest. The test at this speed can better reflect the relationship between the force and the depth. The test depth is from 0 mm to 60 mm/s, and the depth increases by 10 mm between the two tests. It can be seen from the test results that the horizontal force in the two media increases with the increase in depth, and the changes tend to be relatively close. From this set of tests, no significant difference in the forces in the two media is observed. However, the deviation of the measurement results in the wet granular media is significantly greater than in the dry granular media. The standard deviations in the statistical results are 0.03~1.46 for the wet granular media and 0.15~1.21 for the dry granular media. The reason for this phenomenon is because the consistency of the wet granular media is more difficult to guarantee during the test process.

### 2.5. STEP 5: Tests of the Vertical Force

We tested the vertical force by pressing the intruder from the surface of the granular media to a depth of 60 mm at different speeds. The speeds were 10 mm/s, 20 mm/s, and 30 mm/s. Each speed was tested 3 times. The results are shown in Figure 4. For dry granular media, it can be seen from Figure 4a that no obvious difference is reflected in the test results of the vertical force at different speeds. Vertical force is approximately linearly related to the depth at all three speeds.

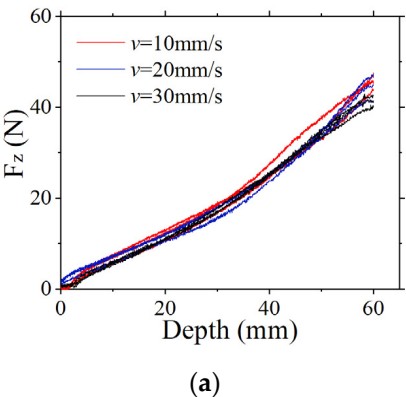
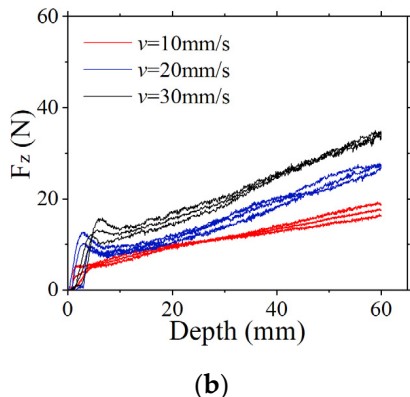

**(a)**                                        **(b)**

**Figure 4.** The results of the vertical force test. (**a**) The test result for GM. Measurements at different speeds did not show significant differences. (**b**) The test result for SWGM. The increasing trend of force with depth is the same at different speeds, but the magnitude of the force is obviously different. The data fluctuation when the intruder entered the SWGM is because there is a process of yielding on the surface of the media.

## 3. Calculation and Verification of the SGRFT

It can be seen from the test results presented in Figures 3 and 4 that two conditions for the RFT of dry GM (force is not sensitive to speed; dimensionless scaling factor for force and depth is nearly the same for all GM tested) are no longer applicable in SGRFT. To predict robot leg resistive force for SWGM ground motion, we extended the RFT of dry GM to the SGRFT relevant to SWGM ground by adding a velocity coefficient and depth coefficient.

### 3.1. SGRFT Method

All the parameters in RFT and SGRFT are defined in Table 2 and shown in Figure 5.

**Table 2.** Definition of the parameters.

| | Parameter | Definition |
|---|---|---|
| 1 | $\beta$ | angle of attack |
| 2 | $\gamma$ | angle of intrusion |
| 3 | $z$ | the distance from the center of intruder to the surface |
| 4 | $A$ | area of intruder's leading face |
| 5 | $S$ | area of leg's leading face |
| 6 | $\sigma_{z,x}$ | vertical and horizontal stress, $\sigma_{z,x} = F_{z,x}/A$ |
| 7 | $F_{z,x}$ | vertical and horizontal force |
| 8 | $\alpha_{z,x}$ | vertical and horizontal stresses per unit depth $\alpha_{z,x} = \sigma_{z,x}/|z|$ |
| 9 | $\alpha_{z,x}^{v}$ | stress (per unit depth) at the speed of $v$ |
| 10 | $\sigma_{z,x}^{v}$ | vertical and horizontal stresses at the speed of $v$ |
| 11 | $\varepsilon_{v}$ | velocity coefficient |
| 12 | $\sigma_{z,x}^{base}$ | the basic stress obtained by testing |

Daniel I. Goldman et al. [10] verified that the intrusion forces in granular media are insensitive to speed when the speed is less than 1 m/s. Based on this condition, the RFT in GM is proposed as shown in (4) and (5). The force is only related to $|z|$, $\beta$, $\gamma$, and the parameters of the GM ($\alpha_{z,x}$). The $\beta$ (angle of attack) and $\gamma$ (angle of intrusion) are obtained from the geometric solution of the leg motion process, $\alpha_{z,x}$ represents vertical and horizontal stresses per unit depth which are measured in GM using a plate element

(intruder), $z$ is the distance from the center of the plate element (intruder) to Surface1 shown in Figure 5, and $|z|$ is the absolute value of $z$. $\sigma_{z,x}$ and $F_{z,x}$ can be calculated by (4) and (5).

$$\sigma_{z,x}(\beta,\ \gamma,\ |z|) = \begin{cases} \alpha_{z,x}(\beta,\ \gamma)\ |z| & z\ <\ 0 \\ 0 & z\ \geq\ 0 \end{cases}' \tag{4}$$

$$F_{z,x} = \int_S \sigma_{z,x}\ (|z|_S,\ \beta_S,\ \gamma_S)dA_S = \int_S \alpha_{z,x}(\beta_S,\ \gamma_S)|z|_S dA_S, \tag{5}$$

$$\sigma_{z,x}(\beta,\ \gamma,\ |z|,\ v) = \begin{cases} \sigma_{z,x}^{base}(\beta,\ \gamma,\ |z|)\ \varepsilon_v & z\ <\ 0 \\ F_w & 0\ \leq\ z\ <\ h\ , \\ 0 & z\ \geq\ h \end{cases} \tag{6}$$

$$\sigma_{z,x}^v(\beta,\ \gamma,\ |z|)\ \approx\ \alpha_{z,x}^v(\beta,\ \gamma)\ |z|, \tag{7}$$

$$F_{z,x} = \int_S \sigma_{z,x}\ (\beta_S,\ \gamma_S,\ |z|_S,\ v)dA_S = \int_S \alpha_{z,x}^v(\beta_S,\ \gamma_S)|z|_S \varepsilon_v dA_S, \tag{8}$$

Through the above research, it can be seen from Figures 3 and 4 that the main variable of RFT in dry GM is no longer applicable in SWGM, which is the $\alpha_{z,x}$ is insensitive to speed. The force of the intruder in the SWGM increases significantly with the increase in speed.

We added the velocity coefficient $\varepsilon_v$ to the RFT of dry GM, obtaining the SGRFT (saturated granular RFT). The test above yields the coefficients ($\varepsilon_v$) for the low-speed range (10 mm/s~50 mm/s). The parameter definition of the intruder in SWGM is shown in Figure 5, and the RFT for SWGM is shown in (6)–(8). $\varepsilon_v$ is the coefficient that the force received by the intruder varies with speed and can be obtained by tests. $\sigma_{z,x}^{base}$ is the basic stress obtained by testing, and other $\sigma_{z,x}$ can be calculated from $\sigma_{z,x}^{base}$ based on (6). $F_w$ is the force generated between the intruder and the water, which can be calculated by (9).

$$F_w = C\rho v^2 S, \tag{9}$$

where $\rho$ is the density of water, $S$ is the projected area of the leading surface, $v$ is the velocity of the leg, and $C$ is the drag coefficient, which is related to the shape of the object and its surface properties.

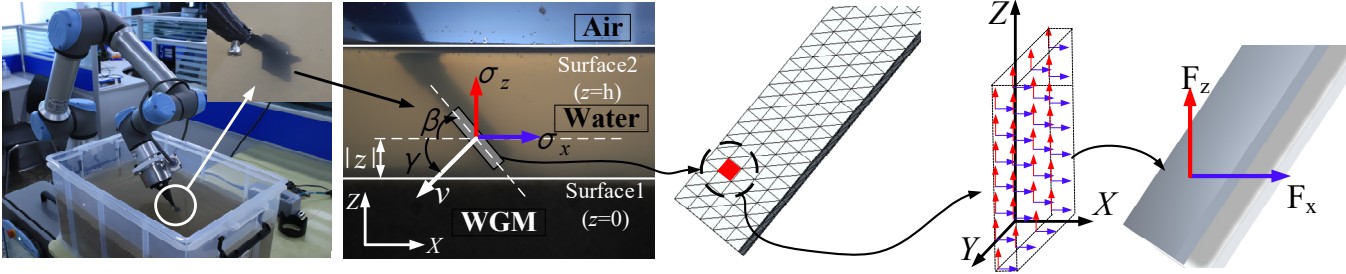

**Figure 5.** Proposed SGRFT methodology. The legs are separated into several tiny flat plate units (invaders), the stress of the intruder under different $\beta$ and $\gamma$ is obtained through testing and calculations, and the stress of each flat plate is superimposed to obtain the force of the leg. The red arrows represent the vertical force, and the blue arrows represent the horizontal force. The size of the box is L × W × H = 550 mm × 380 mm × 350 mm.

The $\varepsilon_v$ is calculated from the tests in Figures 3 and 4b. By analyzing the test data in Figure 3a, force and velocity are approximately linearly related, and the velocity coefficient is the slope of the fitted line, so the velocity coefficient is added as a multiplier in (7).

We performed linear fitting on the test data in Figures 3 and 4b. The intercept is a fixed value, and the change process of force with velocity and depth is mainly related to

the slope, so we took the slope as the coefficient of variation. We obtained the $\alpha_{z,x}^v$ by (10), and $\varepsilon_v$, $\alpha_{z,x}^{10}$, $\alpha_{z,x}^{20}$, and $\alpha_{z,x}^{30}$ were calculated from the test results and are shown in Table 3.

$$\alpha_{z,x}^v = \frac{\sum_{i=1}^n \alpha_{z,x}^v}{n}, \tag{10}$$

Taking the force when the speed is 30 mm/s and the depth is 30 mm in Figure 3 as the base value ($F^{base}$), the remaining force is predicted according to the coefficients in Table 3, and the error map of predicted values is shown in Figure 6. Errors are calculated as (11). $F^{predict}$ is the prediction value, and $F^{test}$ is the test value.

$$\text{Error} = \left| \frac{\left( F^{predict} - F^{test} \right)}{F^{predict}} \right| \tag{11}$$

Among the fourteen prediction results, ten have an error of less than 6%, and only two have an error of greater than 16%. The values with significantly larger errors than other prediction results are concentrated in the position where the speed and depth are the smallest. It can be seen from Figure 4b that the force near the surface of SWGM exhibits local fluctuations and shows a linear trend when the depth is more than 10 mm. This is perhaps why the prediction of forces at minimal depth may be inaccurate.

**Table 3.** Coefficients calculated from test results.

| | Coefficient | Average Value | $R^2$ |
|---|---|---|---|
| 1 | $\varepsilon_v$ | 0.16 | 0.952 |
| 2 | $\alpha_{z,x}^{10}$ | 0.23 | 0.940, 0.977, 0.968 |
| 3 | $\alpha_{z,x}^{20}$ | 0.37 | 0.941, 0.965, 0.961 |
| 4 | $\alpha_{z,x}^{30}$ | 0.47 | 0.933, 0.959, 0.970 |

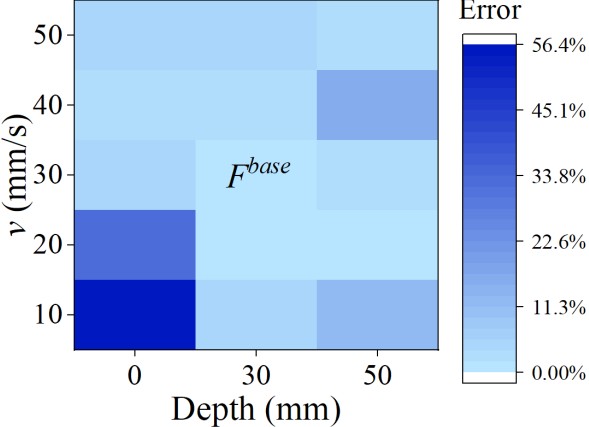

**Figure 6.** Error in the prediction of the result of horizontal force. Each set of data was tested three times. The dark color indicates a large error. More than 70% of the prediction results have an error of less than 6%.

### 3.2. Simulation and Validation for Legs in SWGM

To prove the applicability of SGRFT, we verified the force of three differently shaped legs when moving in SWGM through simulation and experiments, namely, rectangle leg (R-leg), triangle leg (T-leg), and V-shaped leg (V-leg). The projections of the three legs in the direction of motion are all rectangles of 25 mm × 110 mm. The simulation process, test device, and measurement results are shown in Figure 7.

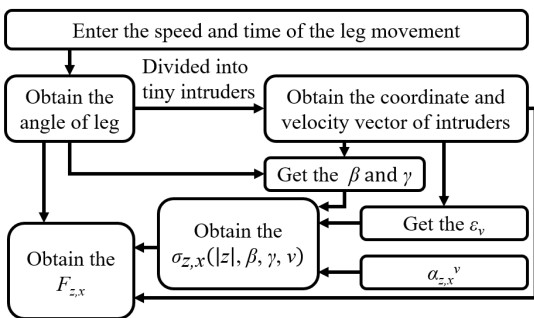

**Figure 7.** Simulation process.

A custom-made waterproof motor drives the legs. The force is measured by a six-dimensional waterproof F/T sensor (ATI Gamma IP68). The sand pit is the one shown in Figure 1. The legs are installed on the output shaft of the motor, and one end of the F/T sensor is installed with the motor. The other end is installed on a fixed frame with a position scale. The fixed frame can adjust the height of the motor output shaft according to the scale to achieve accurate control of the leg subsidence depth during the test.

The subsidence depth of the leg in the simulation and the corresponding test is 50 mm, and the angular velocity of the leg is 2 rpm. According to the comparison of simulation and test results shown in Figure 8, it can be seen that for R-leg and V-leg, the force variation trend in the simulation results obtained according to the SGRFT is consistent with the test results. The peak value of the force in the test results is slightly higher than the simulation results, and the peak position is moved forward. The sand causes it to accumulate in the front of the leg during the moving process and be slightly compacted, so a slight change in the resistance of the legs occurs. It can be observed from the simulation results and test results that when the legs move in the SWGM, the force in the descending and ascending process presents asymmetry, which is also available in the GM. The measured results for the T-leg deviate significantly from the simulation results caused by the obvious force in the 3D space during the motion. At the same time, the SGRFT discussed in this paper and the RFT proposed in [10] is suitable for the 2D space.

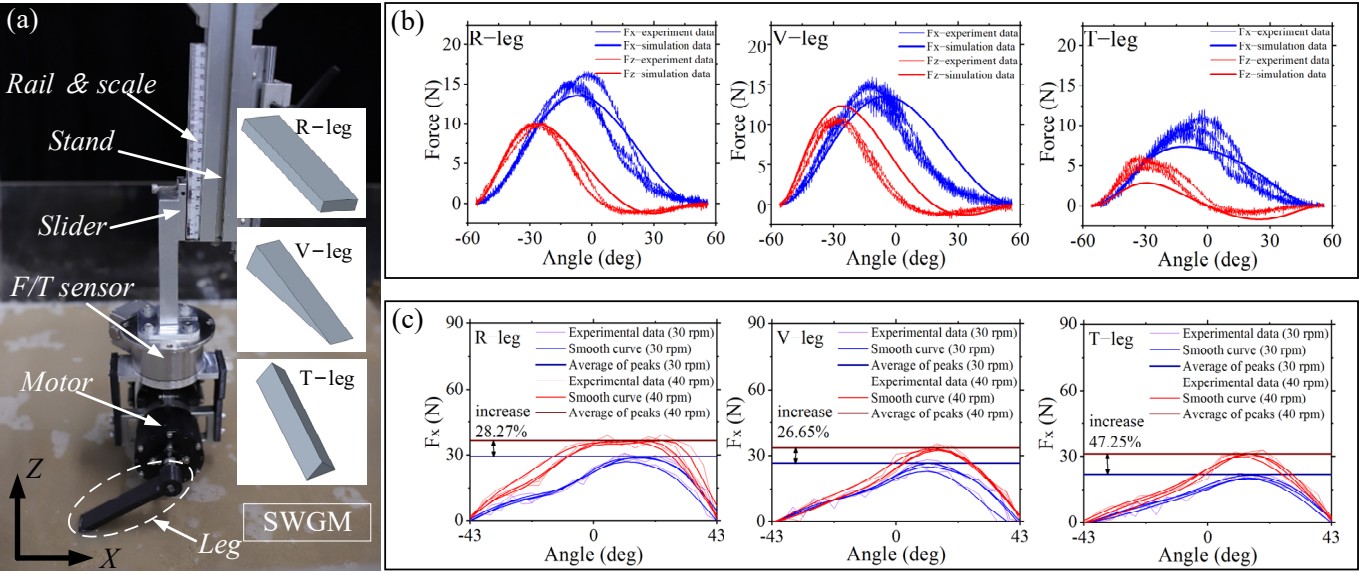

**Figure 8.** Tests and results for three legs. (**a**) The equipment and test scenarios. The frame with scales can adjust the depth that the legs sink into the SWGM, and the F/T sensor and motor are waterproof. For the test of 2 rpm, the motor is equipped with a 50:1 reducer, and when the speed of 30 rpm and 40 rpm is tested, the motor is equipped with a 6.3:1 reducer. (**b**) The testing and simulation results of 2 rpm (depth = 50 mm). (**c**) Horizontal resistive force variation at different speeds.

To verify the effectiveness of $\varepsilon_v$ and $\alpha_{z,x}^v$ in the leg movement process, by using 30 rpm and 40 rpm, we moved three legs in the SWGM at a depth of 30 mm. The measurement results are shown in Figure 8. The value of peaks is the average of the values in the $3°$ interval before and after the angle point at which the extreme force value is located. According to the coefficients in Table 3, it can be predicted that the difference between the horizontal thrusts at the two speeds should be 30.33%. The test results show that the horizontal force changes in the R-leg and V-leg are 28.27% and 26.65%, respectively, which can prove the effectiveness of the prediction coefficient on the leg movement process in 2D space. The prediction results of T-leg have a larger deviation, which is also consistent with the characteristics of the SGRFT simulation results and can only be applied to the prediction of forces generated by 2D space motion.

The insensitivity of force to velocity shown in GM is no longer applicable in SWGM, which is extremely important for the study of the legged locomotion, especially for robots in the natural environment, such as amphibious robots, whose working environment covers GM and SWGM and frequently switches. According to the test results, the similarities and differences between the main force properties of the 2D leg movement process in the SWGM and GM listed in Table 4 can be summarized ($\times$ means negative, and $\sqrt{}$ means affirmative). The characteristic that the force shown in GM is approximately linearly correlated with depth also exists in SWGM, which is of great significance for reducing the measurement of basic data and reducing the workload. Still, the difference is that $\alpha_{z,x}$ is dimensionless and nearly the same for all media tested in GM, but $\alpha_{z,x}^v$ varies significantly with speed in SWGM. In the process of leg movement in GM, the change in force with the direction of symmetrical velocity is asymmetric, which still exists in SWGM.

**Table 4.** Similarities and differences in force laws between GM and SWGM.

| Force Properties | GM | SWGM |
|---|---|---|
| Sensitivity to speed | $\times$ | $\sqrt{}$ |
| Linearly related to depth | $\sqrt{}$ | $\sqrt{}$ |
| The stresses per unit depth $\alpha_{z,x} / \alpha_{z,x}^v$ | Fixed | Linearly related to speed |
| The force distribution is asymmetric | $\sqrt{}$ | $\sqrt{}$ |

## 4. Experiments

We measured the horizontal force of propeller-leg (as shown in Figure 9) motion on the SWGM ground and proved the applicability of $\varepsilon_v$ and $\alpha_{z,x}^v$ for the SWGM, as shown in Figure 10. The comparison experiment of the robot equipped with propeller-legs running on SWGM and GM ground proved that the force laws of the robot's legs in the SWGM and GM ground movement are different.

### 4.1. Test for Propeller-Leg in SWGM

The test method and equipment of the propeller-leg are the same as the tests for the three legs above, and the detailed information is given in Table 5. The method of obtaining the peak value of the force is the same as above. The test results are shown in Figure 10. The variation trend of $F_x$ with depth and speed is in line with the previous research results, but the rotation angle range that can generate effective propulsion at shallow depths is smaller than the theoretical value (the radian of the foot is $0.5\pi$ rad, and the rotation angle that can theoretically generate an effective propulsion range should be $\pi$ rad). This is because the sand cannot be backfilled at shallow depths after being pushed away by the front of the foot, resulting in insufficient sand contact with the back half of the foot. However, the phenomenon improves as the depth increases, and the range of rotation angles that can generate effective propulsion reaches the theoretical value ($\pi$ rad) when the depth reaches 15 mm. This verification shows that the laws of force proposed above are still applicable to other leg forms, except for the simplest shape (rectangular and V-shaped plates).

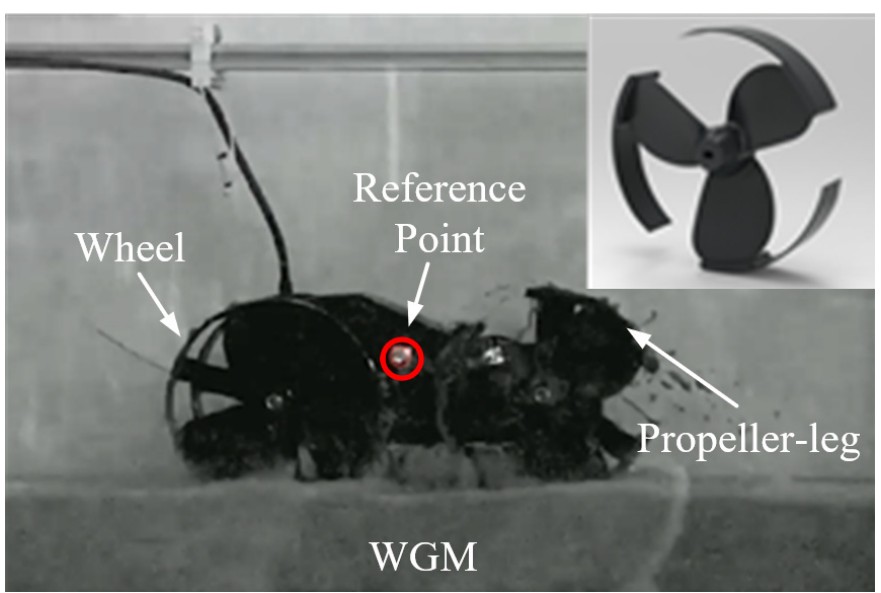

**Figure 9.** A robot equipped with propeller-legs on SWGM ground. The propeller-leg is a kind of propulsion device that was introduced in our previous study [32] for the amphibious robot. Three arc-shaped legs are evenly distributed on a circle with a radius of 110 mm, and the arc is 0.5 rad. The DLT program can recognize the reference point on the body, the frame rate of DLT is 60 Hz, and the number of current points is 1.

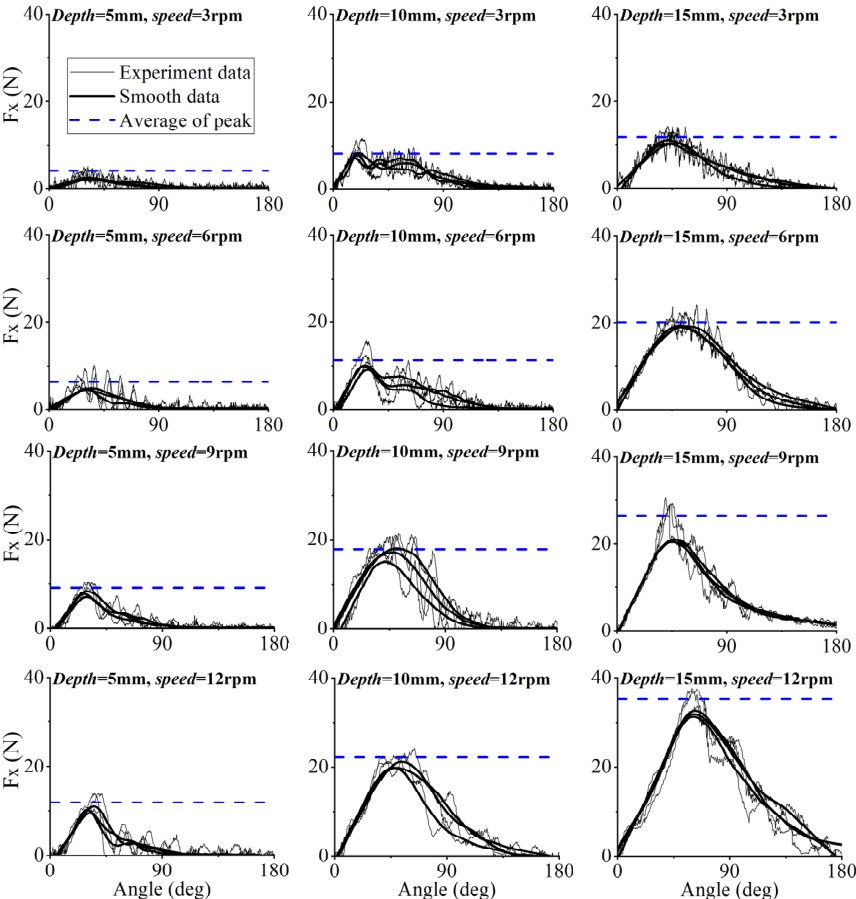

**Figure 10.** The outcome of the propeller-leg test. Representation of results from propeller-leg validation experiments for all combinations of depth and speed. LOWESS smoothing was performed on the test results with a window width of 0.2, and the blue dashed line represents the average peak force.

**Table 5.** Equipment of the test for propeller-leg.

| Equipment | Model |
|---|---|
| F/T sensor | ATI Gamma IP68 |
| Motor | KOLLMORGEN TBM(S)-6013-A |
| Encoder | RLS MRA029BC010DSE00 |
| Driver | G-SOLTWI15/100EE1 |

### 4.2. Setup of the Running Experiment

The robot equipped with propeller-legs is shown in Figure 9. The two propeller-legs at the front of the robot are driven to rotate at the same speed by a waterproof motor and a reducer, and the two wheels at the rear of the robot are passive wheels. An obvious marking point is painted on the body of the robot, which is used to measure the movement speed. The running process of the robot is completed in a transparent cylinder of 1500 mm × 600 mm × 600 mm, and the bottom of the cylinder is laid with GM or SWGM during the corresponding test process.

During the running process on the SWGM ground, the water surface is less than 10 mm higher than the sand surface to ensure that the robot body will not be subjected to the force of water during the movement. The GM/SWGM surface was scraped after each test to ensure the consistency of each test condition. The running process of the robot was filmed with a high-speed camera (Chronos 1.4, Software v0.3.0), and to identify the motion of the robot in the video by direct linear transformation (DLT), we used a freely available software package, DLTdv7 [33]. The program automatically tracks the position of the marker points on the robot body and calculates the average speed of passing the fixed distance (720 mm) as the robot's movement speed. The process of the running experiment and tracking is shown in Figure 11.

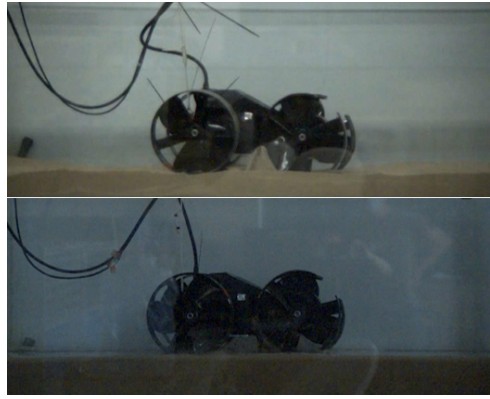

(**a**)

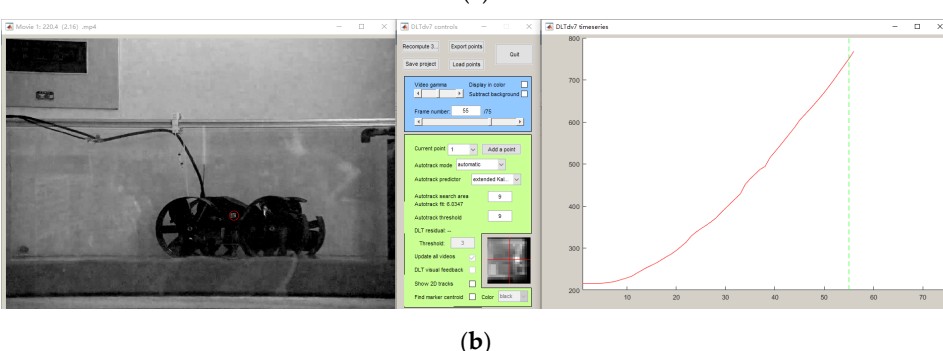

(**b**)

**Figure 11.** The process of the running experiment and tracking. Panel (**a**) shows the running process on GM (above) and SWGM (below) ground. Panel (**b**) shows the scene of the track: the left is the running process, the middle is the parameter setting and operation interface, and the right is the tracking result (the abscissa represents the number of frames, and the ordinate represents the position coordinate).

### 4.3. Results and Comparison of the Running Experiment

Figure 12 shows the results of the running experiment. When the rotation speed of the propeller-leg is less than 60 rpm, the running velocity of the robot on the two kinds of ground is similar. As the speed increases, the horizontal running velocity of the robot in the SWGM is significantly greater than that on the GM ground, and the higher the rotational speed is, the greater the difference in the velocity of the robot. The velocity of the robot in the horizontal direction is mainly determined by the horizontal force between the propeller-leg and the granular media, and the difference in the velocity of the robot can reflect the difference in the horizontal force between the propeller-leg and the ground. It can be seen from the results of the running experiment that the force properties of the leg when the robot is running on the GM and SWGM ground are significantly different and show the sensitivity of the force to the speed on SWGM ground.

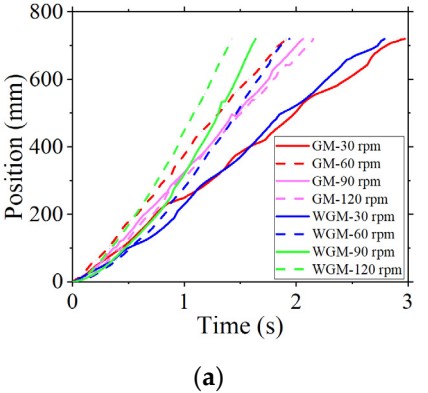
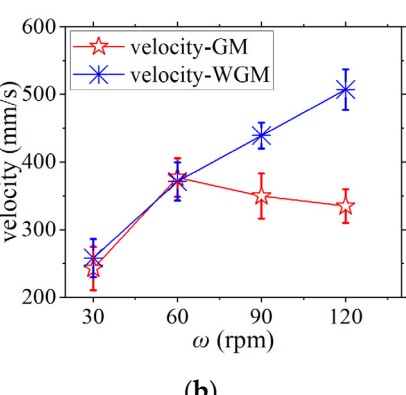

|  (**a**) | (**b**) |

**Figure 12.** The results of the running experiment. (**a**) The trend of the robot's position changes with time. According to the experimental results, the robot performs approximately uniform motion. (**b**) Comparison of two motion processes' velocities. When $\omega$ changes, the change in the robot's velocity on two kinds of ground is different. Error bars are standard deviations, with three trials for each condition.

## 5. Conclusions

We developed a method named SGRFT extended by RFT for predicting the horizontal resistive force of robotic legs when running on SWGM ground. The SWGM ground was formed by mixing sea sand (particle diameter $\leq$ 1 mm) and water, and the saturated wet sand was submerged in water. SGRFT can predict the horizontal propulsion force of the robotic legs at low speed (10 mm/s~50 mm/s) based on a limited set of scaling factors and an additional velocity coefficient calibration. There are some meaningful results in this study which are presented as follows:

(1) The difference in sensitivity to speed was observed by the comparative tests of the leg movement in GM and SWGM. The influence of moving speed on propulsion was much greater in SWGM than in GM. When the speed changes from 10 mm/s to 50 mm/s at the same depth, the change in maximum horizontal force can reach 78.6% in SWGM, but the horizontal force changes in the GM do not exceed 10% under the same measurement conditions.

(2) The SGRFT method was developed by adding the scale factors of velocity coefficient and depth coefficient to RFT. The SGRFT can predict the horizontal propulsion force of legs during low speed (10 mm/s~50 mm/s) motion on the SWGM ground.

(3) The running experiment of the robot equipped with the propeller-leg proved that robotic legs exhibit different force laws on dry GM ground and SWGM ground. The SGRFT can be used as a tool and method to predict the resistive force on the legs of robots working in an amphibious environment.

**Author Contributions:** Conceptualization, X.M. and G.W.; methodology, X.M.; software, X.M. and X.C.; validation, X.M., K.L., J.W. and B.P.; formal analysis, X.M.; investigation, X.M., X.C. and G.W.; resources, X.C. and G.W.; data curation, X.M. and G.W.; writing—original draft preparation, X.M.; writing—review and editing, G.W.; visualization, X.M.; supervision, X.M.; project administration, X.M.; funding acquisition, X.C., G.W. and L.W. All authors have read and agreed to the published version of the manuscript.

**Funding:** This research was funded by The National Natural Science Foundation of China (Grant No. 52001116), The National Natural Science Foundation of Heilongjiang Province, grant number YQ2020E033 and YQ2020E028, in part by The China Postdoctoral Science Foundation funded project under Grant 2018M630343 and The Heilongjiang Postdoctoral Science Foundation funded project under Grant 18649.

**Institutional Review Board Statement:** Not applicable.

**Informed Consent Statement:** Not applicable.

**Data Availability Statement:** All data included in this study are available upon request by contact with the corresponding author.

**Conflicts of Interest:** The authors declare no conflict of interest.

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
