# Peer review of "Granular Resistive Force Theory Extension for Saturated Wet Sand Ground"

_machines, doi:10.3390/machines10090721_

Round 1

Reviewer 1 Report

This paper attempted to develop a resistive force theory for locomotion on wet granular media. I think the work is definitely important and has promise in advancing knowledge in a significant way, but in its current shape, it is not suitable for publication. This is mainly due to the lack of rigor and the writing being overpromising (stating the contribution to be more than it actually offers).

Major comments:

1. Wet granular media is very broad, not just water and sand mixed together. For example, partially wet sand and mud behaves in qualitatively different ways from the saturated sand focused on here. The author made it sound like their work is all encompassing for wet granular media.

2. The resistive force theory is merely a simple expansion of the previous work. As far as I can tell, what the authors did is merely introducing a couple of fitting parameters added to the previous dry resistive force theory, and fit this model to their minimal data (vertical lift and horizontal drag) to obtain the values of these two parameters.

3. In your robot test, why is there no water above the saturated sand as you tested in force experiments? And why do you have water above sand in the first place? It seems that will increase pressure at the sand surface level, which can potentially alter the behavior.

This is in line with my first comment above. Even with the same composition of solid particles, a granular medium at different levels of water (dry, wet, saturated but not submersed in a lot of water, saturated and submersed) can have very different behavior.

The authors are encouraged to read basic mud rheology literature first and not overpromise.

4. The methods and results sections are poorly written and there is a lot I can’t quite understand exactly what the authors did. Please better explain and use figures to define your experimental setup and procedures. I basically got lost in Section 3 after Fig. 5 and can’t understand well what you did here, even though I’m an expert in this exact research area.

5. If your WRFT is so great as claimed, why don’t you apply it to predict performance of the robot experiments like done in Ref. 8?

6. The comprehensive review of literature is nice, but can you better motivate your work beyond just that wet GM hasn’t been studied as much? Just because something is not studied doesn’t mean it’s important and worthwhile to study it. There are billions of questions in this world that haven’t been studied. What makes your topic, approaches, insight, etc. here useful for the broader field? Also, there have been some studies of saturated GM locomotion in animals, e.g., Winter & Hosoi and Gidmark et al.on burrowing, which are relevant and missed.

 7. There are a lot of English grammar issues or improper use of words. The authors should seek professional editing help.

Minor comments:

Also I don’t understand this: the two fitting parameters are both scaling factors, and they are multiplied to yield the overall adjustment to the original model, based on Eqn. 7, 8. Why do you need two? Wouldn’t one be enough?

I can’t see clearly what the differences are between the three legs of different shapes. They look very similar, and I wonder why there is a major difference in force magnitudes.

Line 46: this should be reference 8. All you are applying here are from Ref. 8.

Ref 13 doesn’t seem to be supporting Dynamic RFT mentioned on Line 53.

Don’t list one author “did”. It’s many authors working together. At least XX et al.

You need to say “we did…” in methods. Not just “do…”.

There are results you say you got but didn’t show, like on line 201.

Define what error bars in all figure captions.

Some numbers are missing units.

What is \alpha_10, 0, etc.? What are these numbers?

The box in the Fig. 5 is quite small. Are you sure there is no boundary effect when you measure forces here?

Scissors should be arrows.

Reviewer 2 Report

I liked this paper and I enjoyed reading it. I feel the results are very interesting (unlike in dry granular media where there is a peak rpm for speed, this is not shown in wet granular media).  I could see myself following these methods for my own work, which is why I'm going to make many suggestions to improve the clarity for someone with my own background.  Because the clarity of some portions can be improved. 

Intro feels disorganized, one sentence about octopus, wheels, worms?  Topic sentences will help. 

Please make sure to to include all parameter values. What is alpha generic? is zeta a function of both depth and speed?  I'm not following the calibration between lines 262 and 282.  How many tests were performed?  Where is the reference point (alpha generic?) for fig 6? The error in Fig 6 seems large and should be discussed. 

The discussion seems limited in scope. Will this formulation apply to natural wet granular media where there may be cohesion due to biomaterials?  How accurate is this method?  

I disagree with the conclusion that "it is proved that WRFT can be used ... in  the design process of a legged robot ... " because that was not done in this paper.  The design already existed and this paper characterized it. 

Small Notes

The small insult to NASA's rover feels unnecessary, Spirit rolled 5 miles and opportunity 28miles. Mars is not covered with liquid water and I don't know how this proceedure would apply to Mars soil.

relay and rely have different meanings

often "the" is used incorrectly

capitalization inconsistent : e.g. 2.2 Two calibrations for the Test

line 168: the same quantity of sand rather than same quality?

Figure 4 : add title for a and b

equations (9),(10), (11) have extraneous commas that look like primes.  

262 space before comma

Figure 12 a  - it is hard to distinguish lines.  (b) at first when I looked at legend points I thought they were data points.  solve by boxing legend or connecting points in data series?

why is propeller capitalized?

Reviewer 3 Report

This paper discusses a resistive force model for legged robots wading through wet granular medium. The paper extends resistive force theory developed for granular medium to wet granular medium.

1. The paper should be proof read by a native English speaker to correct grammatical mistakes. As an example, in the first sentence in Abstract,  instead of comma semicolon should be used to join two independent clauses (this pattern continues through out the paper). The second sentence in the Abstract is grammatically incorrect as well.

2. There are long sentences in the paper that need to be divided for clarity. As an example, a single long sentence spans lines 86-90 on page 2.

3. Though the title claims a model of crawling robots, no crawling scenario was considered in the paper. The Abstract mentions running instead of crawling, and the paper contributions listed at the end of Sec. 1 (Introduction) mention motion of the legged robot, but no crawling.

4. Sec. 2.1 (Test Setup) is written as a instruction manual (do this, do that). It should be rewritten to report the experimental procedure followed.

5. Before introducing mathematical equations, such as (1)-(3), first describe their purpose.

6. In Sec. 2.2, it is not clear what is meant by the phrase, "the distance between the two tests is 80mm".

7. The abbrev. FRT used on line 106 (page 3) should be RFT.

8. Table 2 is referenced on page 5, however, the table appears on page 8. Also, it is not clear why three R^2 values are shown in Table 2. Only the average value should be reported.

9. Please explain the sentence (line 211, page 6) "However, the error of the measurement results in the wet granular medium is significantly greater than that in the dry granular medium." Which error is referred here? What is the basis of this claim?

10. Please define |z| in equations (4)-(8). Also, define sigma and alpha variables. Where is the origin of the coordinate system for measuring z?

11. the statement "the relationship between force and |z| is shown in (6)" doesn't appear to be correct.

12.  The velocity coefficient is added as multiplier in (7); what is the reason for doing so?

13. Please explain "we perform linear fitting on test data" (line 268). Which test data is referred here? Which variable is being fit to data? Please show a diagram for linear fitting and justify your model. Showing only the error (Fig. 6) is not sufficient. The error graph seems to show that linear model is inadequate.

14. Section 4 is titled "Experiments." However, experimental results appear earlier in Sec. 2&3. 

15. The WGM model in (7)-(8) was developed for plate intruder. Please justify its applicability to the propeller leg (Sec. 4, Fig. 9). Also, provide parameter values for this application.

Round 2

Reviewer 1 Report

The authors have addressed some of the concerns but not fully. Please do not rush and take the time to address them well.

1.     The title seems more appropriate as “Extending resistive force theory…”, as the main novelty is adding two scaling coefficients.

2.     WRFT is still a bit misleading, as wet can be saturated or not saturated. Maybe SGRFT (saturated granular RFT)?

3.     I still think there is a major opportunity to bring this paper to a significantly higher level, if the authors can apply the techniques (using a dynamic simulation engine, which is common these days, such as ADAMS, COMSOL, etc.) to predict the locomotion of the robot and demonstrate how well the extended RFT works in predicting locomotion.

4.     However, if the authors choose not to attempt to predict robot locomotion using the extended RFT, then you cannot claim that the robot experiments proves that the extended RFT can predict the motion of the robot running on SWGM. This is claimed several times, in abstract, introduction, and conclusion.

5.     Sec. 3 can still use a clear summary of what you are doing to the original RFT at a high level (without going into technical detail and terminology) at the beginning, to clearly explain how this work extends the dry GM RFT.

6.     It’s still not clear how all the parameters are defined for Eqns. 4-11. Please add a schematic to define them.

7.     You gave values specifically for a few velocities in Table 2. But a RFT should generally apply to other speeds within the range tested. Can you somehow do this (such as the polynomial fitting in Ref. 10 supplementary material)? The same goes for the text around Eqn.  9-11. Can you make it general?

8.     Can you show how the errors in Fig. 6 are defined in Fig. 8 or Fig. 4 or somewhere? Basically, having an error map like this suggests that you are quantifying how well the model captures data. Can you visualize this somehow? And please give a clear formula defining how the errors are calculated.

9.     Error bars are still not defined in figure captions. Are they standard deviations? Standard errors? Please also add sample size (how many trials each condition) to figure caption.

10.  Author names should be consistent. You sometimes use first name, sometimes last name, some times full name, sometimes acronyms in names. Just use last name et al.

11.  I’m not sure if it’s straightforward to calculate F_w from Navier-Stoke’s equation. Is there a derived form (lift and drag equations in fluids) you can replace this statement with?

12.  The photos of the three legs are still difficult to see. Can you take side view images to clearly show differences?

13.  Just because Ref. 10 found that a distance greater than 30 mm from the boundary is sufficient to minimize boundary effect does not mean that’s necessarily true here. I don’t question you’ve checked this, but you need to state it correctly, that above what distance you found not significant increase in force as the intruder approaches the boundary.

14.  You clearly have data, so the data availability statement is not correct.

Reviewer 2 Report

Thank you for the edits.  Some English and grammar is still awkward but content is clarified 

Author Response

Thank you for your careful review. We have rechecked the manuscript and corrected the grammar.
